# *AaSlt2* Is Required for Vegetative Growth, Stress Adaption, Infection Structure Formation, and Virulence in *Alternaria alternata*

**DOI:** 10.3390/jof10110774

**Published:** 2024-11-07

**Authors:** Qianqian Jiang, Tiaolan Wang, Yongcai Li, Yang Bi, Miao Zhang, Xiaojing Wang, Dov B. Prusky

**Affiliations:** 1College of Food Science and Engineering, Gansu Agricultural University, Lanzhou 730070, China; 2College of Applied Technology, Gansu Agricultural University, Lanzhou 730070, China; 3Department of Postharvest Science of Fresh Produce, Agricultural Research Organization, Rishon LeZion 7505101, Israel

**Keywords:** *AaSlt2*, stress adaption, cell wall integrity, infection structure formation, virulence

## Abstract

Slt2 is an important component of the Slt2-MAPK pathway and plays critical regulatory roles in growth, cell wall integrity, melanin biosynthesis, and pathogenicity of plant fungi. *AaSlt2*, an ortholog of the *Saccharomyces cerevisiae Slt2* gene, was identified from *A. alternata* in this study, and its function was clarified by knockout of the gene. The Δ*AaSlt2* strain of *A. alternata* was found to be defective in spore morphology, vegetative growth, and sporulation. Analysis of gene expression showed that expression of the *AaSlt2* gene was significantly up-regulated during infection structure formation of *A. alternata* on hydrophobic and pear wax extract-coated surfaces. Further tests on onion epidermis confirmed that spore germination was reduced in the Δ*AaSlt2* strain, together with decreased formation of appressorium and infection hyphae. Moreover, the Δ*AaSlt2* strain was sensitive to cell wall inhibitors, and showed significantly reduced virulence on pear fruit. Furthermore, cell wall degradation enzyme (CWDE) activities, melanin accumulation, and toxin biosynthesis were significantly lower in the Δ*AaSlt2* strain. Overall, the findings demonstrate the critical involvement of *AaSlt2* in growth regulation, stress adaptation, infection structure formation, and virulence in *A. alternata*.

## 1. Introduction

*Alternaria alternata* is a latent pathogen responsible for black spot disease in pear [1], jujube [2], sweet cherry [3], and various other fruits and vegetables globally, resulting in significant economic losses. Furthermore, *A. alternata* can generate AK, ACT, and other poisons, posing a risk to human health [4]. The predominant strategy for managing black spot involves the application of chemical fungicides; nevertheless, this approach has resulted in drug resistance and pesticide residues, raising food safety concerns, and contributing to environmental issues [5]. Thus, a thorough comprehension of the response and recognition mechanisms on host surfaces, together with the subsequent pathogenic processes of plant fungus, is essential for formulating targeted disease management methods.

Mitogen-activated protein kinases (MAPK) are part of the serine/threonine protein kinase family, comprising MAP kinase kinase kinase (MAPKKK), MAP kinase kinase (MAPKK), and MAP kinase (MAPK) [6]. MAPK cascade pathways are extremely conserved among eukaryotes, typically activated by various environmental stimuli, subsequently transmitting signals to downstream kinases and transcription factors, ultimately regulating the transcription levels of pertinent genes [7]. In plant fungus, MAPK pathways govern growth, secondary metabolism, stress adaptation, and pathogenicity [8,9]. Cao et al. [10] demonstrated that MAPK cascades provide crucial regulatory functions in proliferation, sporulation, and pathogenicity in *Cytospora chrysosperma*. In *Colletotrichum gloeosporioides*, the disruption of *CgMK1* resulted in the loss of the ability to generate appressoria [11]. It has been reported that five conservative MAPK signaling pathways in *Saccharomyces cerevisiae* are regulated by Hog1 (high osmolarity response), Slt2 (the cell wall repair and integrity), Smk1 (sporulation-specific mitogen-activated protein kinase), Fus3 (the pheromone response), and Kss1 (the pseudohyphal and invasive growth upon nutrient deprivation), which are involved in osmoregulation, cell wall integrity, conidial assembly, pheromone response, and mating, respectively [12]. Notably, only three MAPK signaling cascades-Fus3/Kss1-MAPK, Hog1-MAPK, and Slt2-MAPK-have been discovered in filamentous fungi. Extensive research has demonstrated that the Fus3/Kss1-MAPK pathway regulates reproduction, pathogenicity, and conidial development [13,14,15], while the Hog1-MAPK route is implicated in osmotic stress, conidial development, and the virulence of several pathogens [16,17]. Our prior research validated that the AaHog1-MAPK cascade pathway is implicated in growth, infection structures formation, melanin accumulation, and virulence by pharmacological and molecular biology techniques in *A. alternata* [18].

The Slt2-MAPK pathway, also known as the cell wall integrity (CWI)-MAPK pathway, is a crucial component of MAPK cascades, significantly influencing conidia, oxidative stress, cell wall integrity, and the pathogenicity of pathogens [19]. Slt2-MAPK is crucial for sustaining the cell wall integrity in *Beauveria bassiana* [20]; analogous regulatory models have also been documented in *Phytophthora sojae* [21] and *Ganoderma lucidum* [22]. Slt2 is a downstream component of the Slt2-MAPK signaling pathway and is intricately associated with growth, development, cell wall integrity, and virulence [22,23]. The deletion of the *AflSlt2* gene in *Aspergillus flavus* led to diminished virulence and atypical infection hyphae [24]. Moreover, Spada et al. [25] used silencing methods to show that *Bmp3* influenced the development and pathogenicity of *Botrytis cinerea*.

Pharmacological experiments utilizing the MAPK pathway-specific inhibitor SB203580 (pyridoimidazoles, a specific inhibitor of the MAPK pathway) have previously proven that MAPK cascade pathways are involved in the development, infection structure formation, and pathogenicity of *A. alternata* [26]; nevertheless, the precise molecular processes remain ambiguous. This present research elucidates the roles of *AaSlt2* in the vegetative growth, conidia development, stress adaptation, infection structure formation, and virulence of *A. alternata* via the developed *AaSlt2* deletion mutant and complemented strains. These findings will contribute to the theoretical foundation for enhancing better disease control techniques.

## 2. Materials and Methods

### 2.1. Fungal Strains and Growth Conditions

The *A. alternata* JT-03 strain was prepared as a spore suspension (10^5^ spores/mL), combined with 80% glycerol (Solarbio, Beijing, China) (3:1, *v*/*v*), and preserved at −80 °C. The wild-type (WT) strain was cultured on potato dextrose agar (PDA) at 28 °C and then configured for conidial suspension (10^5^ spores/mL).

### 2.2. Bioinformatics Analysis

The AaSlt2 (XP_018391030.1) in *A. alternata* JT-03 was identified from *A. alternata* (*SRC1lrK2f*, taxon: 5599) utilizing the blastp algorithm with Slt2 (NP_011895.1) from *S. cerevisiae*. Amino acid sequences of Slt2 in other fungi were obtained from the NCBI database, and amino acid sequence alignments were conducted using DNAMAN 6.0. Conserved domains are a series of functional regions in proteins that remain relatively unchanged during evolution. By identifying these conserved domains in protein sequences, the function and evolution of proteins can be better understood. We predicted the conserved domains of AaSlt2 using Conserved Domains (https://www.ncbi.nlm.nih.gov/Structure/cdd, accessed on 6 January 2021).

### 2.3. Deletion and Complementation of AaSlt2

The *AaSlt2* knock-out vector was acquired by a homologous recombination method (Appendix A). The upstream and downstream portions (about 1 kb) of the *AaSlt2* gene were amplified from WT genomic DNA using the primer pairs *AaSlt2*-up/down-F/R (Appendix A), and subsequently cloned into pCHPH upstream and downstream of *hph*, respectively. The fusion cassette comprising the upstream and downstream flanking sequences along with *hph* was subsequently introduced into the *A. alternata* JT-03 by employing *Agrobacterium tumefaciens*-mediated transformation (*AtMT*). The transformants were selected from induction medium plates with 200 μM acetosyringone (Solarbio, Beijing, China) and validated using PDA plates with carbenicillin (Solarbio, Beijing, China) (500 μg/mL) and hygromycin B (Solarbio, Beijing, China) (250 μg/mL). Additionally, deletion strains (Δ*AaSlt2*) were identified by PCR (Appendix A) and qPCR (Appendix A) using the primers listed in Appendix A, respectively.

To achieve complementation, the fragment encompassing the CDS sequences of *AaSlt2* was PCR-amplified (Appendix A) utilizing cDNA from the WT with the primer pairs listed in the Appendix A. Subsequently, it was included into the pC-NEO-NGFP vector. The recombinant plasmids were introduced into the Δ*AaSlt2* strain using *AtMT*-mediated transformation. The selected transformants were confirmed using PDA plates supplemented with containing kanamycin sulfate (Solarbio, Beijing, China) (50 μg/mL) and G418 (Solarbio, Beijing, China) (250 μg/mL). Additionally, complementation strains (Δ*AaSlt2*-C) were discovered by GFP fluorescence detection (not displayed) and PCR. Appendix A shows the list of primers used.

### 2.4. Determination of Spore Morphology, Vegetative Growth and Sporulation

To examine spore morphology, 20 µL spore suspensions of WT, Δ*AaSlt2*, and Δ*AaSlt2*-C were incubated on a slide, and viewed under a microscope. To observe vegetative growth, 2 µL spore suspensions of WT, Δ*AaSlt2*, and Δ*AaSlt2*-C were cultured on PDA at 28 °C for 5 days, with subsequent assessment of colony morphology and diameter. To ascertain of sporulation, 2 µL spore suspensions of WT, Δ*AaSlt2*, and Δ*AaSlt2*-C were incubated on PDA at 28 °C for 5 days. They were then collected and resuspended in conidia and enumerated under a microscope using a hemocytometer. The experiment was performed independently three times.

### 2.5. Gene Expression Analysis

Treatment 1: hydrophobic film (Univ-bio, Shanghai, China) was spread on the slide; treatment 2: hydrophobic film was spread on the slide, and then pear wax was evenly coated on the hydrophobic film. First, 20 µL spore suspensions (10^5^ spores/mL) of WT were incubated on hydrophobic and pear wax extract-coated hydrophobic film. After 2, 4, 6 and 8 h of incubation, the spore slurry was harvested, centrifuged, and precipitated for further RNA extraction. RNA and cDNA were acquired using the manufacturer’s protocol (Vazyme, Nanjing, China). The primer pairs were listed in Appendix A. The gene relative expression level was calculated using the 2^−ΔΔCT^ method following the methodology of Livak and Schmittgen [27].

### 2.6. Infection Structure Formation Assays

The onion epidermis experiments were conducted according to the methodology of Tang et al. [28]. The inner membrane of the onion epidermis was excised and sectioned into 20 × 20 mm squares, which were subsequently positioned on slides (intact onion epidermis). Pear wax, dissolved in chloroform, was uniformly applied to the inner membrane of the onion with an applicator (pear wax extract-coated onion epidermis). Subsequently, 20 µL spore suspension of WT, Δ*AaSlt2*, and Δ*AaSlt2*-C was applied to intact onion epidermis (θ1) and pear wax extract-coated onion epidermis (θ2). The rates of spore germination, appressorium formation, and infection hyphae formation were assessed after 2, 4, 6 and 8 h of incubation. The experiment was performed independently three times.

### 2.7. Stresses Adaption Assays

Spore suspension of WT, Δ*AaSlt2,* and Δ*AaSlt2*-C was prepared as previously described. To evaluate the response of Δ*AaSlt2* strains to cell wall stress and oxidative stress, PDA medium was formulated with congo red (Solarbio, Beijing, China) (100 μM), sodium dodecyl sulfate (Solarbio, Beijing, China) (SDS) (0.01%) and H_2_O_2_ (Solarbio, Beijing, China) (3 mM). Then, 2 µL spore suspensions of WT, Δ*AaSlt2*, and Δ*AaSlt2*-C were correspondingly deposited in the media. PDA medium devoid of such additives was designated as control. Colony morphology and diameter were assessed for 5 days at 28 °C. The experiments were conducted three times.

### 2.8. Assays for Pathogenicity

Pear fruits were sanitized with 0.1% of sodium hypochlorite (Solarbio, Beijing, China) and rinsed with tap water. After the surplus water was removed, the pear fruits were punctured with sterilized nails at three equidistant points around the equator of each pear. Subsequently, 20 µL of spore suspension from WT, Δ*AaSlt2,* and Δ*AaSlt2*-C were injected into each wound, respectively. The pear fruits were placed in a plastic bag at 28 °C. Following incubation periods of 3, 5, 7, 9 and 11 days, the diameters of the lesions were measured. Each treatment consisted of nine pear fruits.

### 2.9. Assessment of Cell Wall Degrading Enzyme Activity

The activity of the cell wall degrading enzymes (CWDEs) was monitored using method by Jia et al. [29]. Preparation of crude enzyme solution: the mycelium of WT, Δ*AaSlt2*, and Δ*AaSlt2*-C was inoculated into 15 mL of PDB and incubated at 28 °C with shaking at 220 rpm for 4 days. Following filtration through four layers of sterile gauze and subsequent rinsing with sterile distilled water, the mycelium was transferred to a medium containing pectinase (Solarbio, Beijing, China) and cellulase (Solarbio, Beijing, China) for a duration of 0 to 9 days. Following incubation for 1, 3, 5, 7 and 9 days, the samples were filtered and centrifuged at 4 °C and 12,000 rpm.

Assessment of polygalacturonase (PG) and pectinmethylgalacturonase (PMG) enzyme activities: addition of 0.5 mL of crude enzyme solution and inactivated enzyme solution to a pretreatment mixture consisting of 50 mM acetic acid-sodium acetate buffer, 10 g/L PG (Solarbio, Beijing, China) and PMG (Solarbio, Beijing, China) substrate, followed by incubation in a 37 °C water bath for 5 min, and subsequently for 1 h at the same temperature. Next, 1.5 mL of 3,5-dinitrosalicylic acid (Solarbio, Beijing, China) was added promptly, followed by boiling for 5 min, and then rapidly cooled to room temperature. The absorbance was recorded at 540 nm. The enzymatic activity of PG and PMG were assessed by measuring the quantity of reducing sugar liberated during the enzymatic reaction. The enzyme activity unit is defined as the quantity of enzyme necessary to facilitate the liberation of 1 μg of reducing sugar from the substrate by 1 mg of enzyme protein per hour at 37 °C. The experiments were conducted three times. The equation for determining enzyme activity is as follows: Enzyme activity (U/mg) = (C × N × OD)/T, where C represents substrate concentration, N denotes substrate amount, OD indicates absorbance value, and T signifies time. 

Assessment of Cx (cellulase) and β-glucosidase activity: After altering PG and PMG substrates with Cx (Solarbio, Beijing, China) and β-glucosidase (Solarbio, Beijing, China) substrates, the remainder of the operation was identical to the aforementioned steps.

### 2.10. Measurement of Melanin

The melanin content was measured according to the method of Zhang et al. [30]. For standard curve preparation, melanin standards were created at several concentrations (0, 10, 20, 30, 40, 50, 60 mg/L), and the absorbance values of these solutions at 400 nm were recorded to construct a standard curve. Spore suspensions (2 µL) from the WT, Δ*AaSlt2*, and Δ*AaSlt2*-C strains were grown on PDA, covered with sterile cellophane sheets for five days at 28 °C. Weighed 0.25 g of mycelium was boiled in 1 M NaOH (Solarbio, Beijing, China). After cooling, the mixture was filtered and pH adjusted to 2.0. Subsequently, the crude melanin solution was obtained using centrifugation.

To purify and measure the melanin, 5 mL of 7 M HCl (Solarbio, Beijing, China) was added to the crude solution and heated in a water bath for 2 h. Subsequent to cooling, the mixture was centrifuged at 10,000× *g* for 15 min, followed by dissolution of the precipitate in 1 M NaOH, adjusting the pH to 2 using 7 M HCl, and centrifugation again at 10,000× *g* for 15 min. The purified melanin was obtained by repeating these steps three times. Following the dissolution of the precipitate with NaOH, the absorbance of the solution at 400 nm was quantified using an ultraviolet spectrophotometer. NaOH (1 M) served as the control solution. The formula for determining melanin content was y = x + 0.111/0.791 (where x is the absorbance value).

### 2.11. Measurement of Toxins

First, 2 µL spore suspensions of WT, Δ*AaSlt2*, and Δ*AaSlt2*-C strains were cultured on PDA, covered with sterile cellophane sheets for 5 days at 28 °C. Subsequently, 0.5 g of mycelium was weighed as sample. Toxins were extracted using an acetonitrile-water (80:20 *v*/*v*) solution with 0.3% formic acid (Solarbio, Beijing, China) and subjected to agitation for 30 min at 150 rpm. Next, 0.25 g of 0.04 M MgSO_4_ (Solarbio, Beijing, China) was added, followed by shock, centrifugation, and filtration of the supernatant using a 0.22 μm filtration membrane. This was then analyzed using an HPLC system. The separation and qualitative analysis of altenuene (ALT) and tentoxin (TEN) were conducted utilizing an Agilent 1260 Q-TOF mass spectrometer (Agilent, Santa Clara, CA, USA) fitted with an electrospray ionization (ESI) source. The HPLC parameters pertained to the testing methodology established by Xu et al. [31]. ALT and TEN standard products (HPLC > 98.0%) were acquired from Pribolab (Qingdao, China).

### 2.12. Statistical Analysis

Statistical analysis was performed using analysis of variance (ANOVA) in SPSS 18.0, with significant differences reported as 95% confidence intervals utilizing Duncan’s multiple range test. Pearson correlations was used to quantify the relationships among factors. The data are presented as the mean ± standard deviation of triplicate measurements. 

## 3. Results

### 3.1. Identification and Bioinformatics Analysis of AaSlt2

*AaSlt2* was anticipated to encode proteins with 416 amino acid residues. The conserved domain analysis revealed that the AaSlt2-encoded protein possesses a characteristic STKc_MPK1 domain, which is part of the PKc_like superfamily (catalytic domain of protein kinases) (Appendix A). The PKc_like superfamily mostly consists of the catalytic domains of serine/threonine-specific and tyrosine-specific protein kinases. It also encompasses RIO kinases, which are characteristic serine protein kinases, aminoglycoside phosphotransferases, and choline kinases. These proteins catalyze the transfer of the gamma-phosphoryl group from ATP to hydroxyl groups in specific substrates such as serine, threonine, or tyrosine residues of proteins.

Amino acid sequence alignment revealed that AaSlt2 is identical to other fungal Slt2 proteins, possessing a ‘TEY’ phosphorylation motif and characteristic sequences of the Ser/Thr protein kinase ‘VlHRDLKPGNLLV’ (Figure 1). To explain the functions of AaSlt2 in *A. alternata*, deletion mutant (Δ*AaSlt2*) and complemented strains (Δ*AaSlt2*-C) were produced.

### 3.2. AaSlt2 Is Involved in Spore Morphology Development, Vegetative Growth and Sporulation of A. alternata

Microscopic examination indicated that the wild-type spore was elliptical and developed a germ tube at the terminal tip, while the Δ*AaSlt2* spore was elongated and produced a germ tube from the lateral side. Moreover, Δ*AaSlt2* exhibited a greater number of septa compared to the WT. The spores of the Δ*AaSlt2*-C strain were equivalent to those of the WT strain (Figure 2A). The deletion of *AaSlt2* resulted in a substantial drop in vegetative growth after 5 days of induction, demonstrating a 35% reduction in colony diameter, relative to the WT. The colony color of Δ*AaSlt2* was drastically reduced compared to the WT, but Δ*AaSlt2*-C restored normal growth morphology (Figure 2B,C). The sporulation of Δ*AaSlt2* was decreased by 68.13% (*p* < 0.05) relative to the WT (Figure 2D). The results indicated that the deletion of *AaSlt2* is non-lethal in *A. alternata*; however, it induced abnormalities in spore morphology, vegetative growth, and sporulation.

### 3.3. AaSlt2 Is Essential for Infection Structure Formation of A. alternata

#### 3.3.1. The Expression of the *AaSlt2* Gene Is Up-Regulated During Infection Structure Formation

qRT-PCR analysis indicated that the expression levels of *AaSlt2* were considerably up-regulated during infection structure formation of *A. alternata* under pear wax extract-coated surface. The expression levels of *AaSlt2* peaked at the germ tube elongation stage (4 h) on both hydrophobic and pear wax extract-coated surfaces, exhibiting increases of up to 21.24 and 54.98 times, respectively, compared to the spore germination stage (2 h). During the germ tube elongation stage (4 h), the expression levels of the *AaSlt2* were elevated on the pear wax extract-coated surface, being 1.59 times greater than on the hydrophobic surface (Figure 3), suggesting that *AaSlt2* is essential for pear wax stimulated infection structure formation of *A. alternata*.

#### 3.3.2. *AaSlt2* Regulates the Infection Structure Formation of *A. alternata* on Onion Epidermis

The process of infection structural differentiation of *A. alternata* consists of four stages. Firstly, the spore germination extends the germ tube, which continues to elongate at the tip to form an appressorium. Over time, the tip of the appressorium extends into a bacterial filament (infection hyphae), which will infests the host, causing infection. To understand the roles of *AaSlt2* in the infection structure formation of *A. alternata*, onion epidermis was used to simulate pear peel. The results indicated that pear wax extract-coated onion epidermis (θ2) markedly improved spore germination, appressorium formation, and infection hyphae formation in the WT, Δ*AaSlt2,* and Δ*AaSlt2*-C strains compared to intact onion epidermis (θ1). At the spore germination stage (2 h), more than 74% of the Δ*AaSlt2* conidia did not germinate on pear wax extract-coated onion epidermis (θ2) (*p* < 0.05) (Figure 4A). Appressorium and infection hyphae formation were induced at 4 h and 6 h, respectively, on pear wax extract-coated onion epidermis (θ2) (Figure 4B,C). The appressorium formation rate of the Δ*AaSlt2* strain on the pear wax extract-coated onion epidermis (θ2) was dramatically reduced by 75% at 6 h compared to the WT (*p* < 0.05) (Figure 4B). After 8 h of incubation, Δ*AaSlt2* showed a 92% reduction in infection hyphae production on pear wax extract-coated onion epidermis (θ2) compared to WT (*p* < 0.05) (Figure 4C), indicating that *AaSlt2* is important for infection structure formation in *A. alternata* when recognizing and responding to pear wax.

### 3.4. AaSlt2 Is Susceptible to Cell Wall Inhibitor Agents and Oxidative Stressors

Slt2, a crucial enzyme in MAPK pathways that responds to oxidative stress and cell wall integrity, has unknown mechanisms in *A. alternata*. As illustrated in Figure 5, in the presence of congo red, SDS, and H_2_O_2_, growth inhibition was more pronounced in Δ*AaSlt2* compared to WT. The growth of Δ*AaSlt2* was decreased by 56.38% when grown on PDA with 3 mM H_2_O_2_. With the addition of 100 μM congo red, the Δ*AaSlt2* only formed colonies with limited growth compared to WT, and the inhibitory rates were 68.19%, indicating that *AaSlt2* is involved in cell wall integrity and oxidative stress adaption in *A. alternata*.

### 3.5. AaSlt2 Is Essential for the Pathogenicity of A. alternata on Pear Fruit

To evaluate the role of *AaSlt2* in the pathogenicity of *A. alternata*, 20 µL spore suspensions of WT, Δ*AaSlt2*, and Δ*AaSlt2*-C were inoculated into damaged pear fruit. The results indicated that WT, Δ*AaSlt2*, and Δ*AaSlt2*-C developed black spots on pears within 1–3 days; however, with extended incubation, the lesion diameter on pears infected with the Δ*AaSlt2* strain exhibited minimal expansion (Figure 6A). The lesion diameters on pear fruit inoculated with the Δ*AaSlt2* strain diminished by 57.07% after 3 days of inoculation compared to the WT (Figure 6B), demonstrating that *AaSlt2* is essential for the pathogenicity of *A. alternata*.

### 3.6. Deletion of AaSlt2 Results in a Decrease of CWDEs Activity

To further investigate the impact of *AaSlt2* on the pathogenicity of *A. alternata*, the activity of CWDEs (PG, PMG, Cx, and β-glucosidase) was assessed. As illustrated in Figure 7, the activities of PG, PMG, Cx, and β-glucosidase were diminished in the Δ*AaSlt2* strain compared to the WT. PG activity increased following 1 day of inoculation, subsequently declining sharply, with the PG activity in the Δ*AaSlt2* strain exhibiting a substantial reduction of 76.27% after 5 days of incubation (*p* < 0.05) in comparison to the WT (Figure 7A). The PMG activity of the Δ*AaSlt2* strain decreased by 47.70% after 1 day of incubation compared to the WT (*p* < 0.05) (Figure 7B). Likewise, Cx and β-glucosidase activity diminished by 72.55% and 47.33% after 5 days of inoculation compared to WT, respectively (*p* < 0.05) (Figure 7C,D), suggesting that *AaSlt2* influences the pathogenicity of *A. alternata* by modulating the activities of CWDEs.

### 3.7. Deletion of AaSlt2 Results in a Reduction of Melanin Accumulation and Toxin Production of A. alternata

To investigate the potential influence of *AaSlt2* on the colony pigmentation of *A. alternata*, we investigated the accumulation of melanin. The melanin content of Δ*AaSlt2* was diminished by 41.45% relative to that of the WT (*p* < 0.05) and was largely reinstated in the Δ*AaSlt2*-C (Figure 8A). ALT and TEN are significant toxins in *A. alternata*. The ALT and TEN levels of Δ*AaSlt2* were reduced by 45.21% and 90.96%, respectively, compared to the WT (*p* < 0.05), respectively (Figure 8B). These results indicated that *AaSlt2* regulated melanin accumulation and toxin biosynthesis in *A. alternata*.

## 4. Discussion

MAPK cascades are conserved signaling pathways in filamentous fungi, implicated in growth, secondary metabolism, stress adaptation, cell wall integrity, and virulence [32,33]. A considerable amount of research has been conducted to elucidate the mechanism of action of these conserved MAPK cascades in various fungi. *BbSte12* and *Bbmpk1* are implicated in the proliferation, oxidative stress response, and hyphal differentiation of *B. bassiana* [34]. Slt2-MAPK, a component of the MAPK cascade pathways, is crucial in regulating cell differentiation, cell wall integrity, conidial germination, and pathogenicity [35,36]. Slt2 is a crucial protein in the Slt2-MAPK pathway, which regulates cellular responses to host and environmental stimuli in fungal infections [37].

The *AaSlt2* gene was identified from *A. alternata* JT-03 in our research. The homology study of amino acid sequences indicated that Slt2 exhibited a homology of 69.28% in *A. alternata, S. cerevisiae*, and *Colletotrichum orchidophilum*. Analysis of the amino acid sequence indicated that the protein encoded by *AaSlt2* possesses the phosphorylation motif ‘TEY’ and the distinctive sequence ‘VlHRDLKPGNLLV’ (Figure 1). MAPK cascades are part of the Ser/Thr protein kinase family, characterized by a conserved threonine-x-tyrosine (TXY) motif across many fungi, suggesting that Slt2 has been highly conserved throughout evolution [38].

In this investigation, we described the *AaSlt2* deletion mutant (Δ*AaSlt2*) and the complementary strain (Δ*AaSlt2*-C) derived from *A. alternata* JT-03. Phenotypic analysis indicated that the deletion of *AaSlt2* influenced spore morphology, vegetative growth, and sporulation in *A. alternata* (Figure 2). In *Mycosphaerella graminicola*, the deletion of *Slt2* resulted in considerable deficiencies in conidia [39]. These results are consistent with a prior publication on *Arthrobotrys oligospora* [40]. In Citrus *A. alternata*, the *Slt2* deletion mutant exhibits significant impairments in colony proliferation, conidial germination, and aerial hyphal growth, with conidia sizes exceeding those of the wild type. Bashi et al. [41] revealed that the loss of *smk3* greatly impacted sclerotium formation and aerial hyphal growth in *Sclerotinia sclerotiorum*, demonstrating that *Slt2* is crucial for spore morphology, vegetative growth, and sporulation across several fungal species.

Fungal pathogens can detect and react to the physicochemical properties of plant surfaces, such as hydrophobicity and the composition of cutin and wax, subsequently activating G proteins, cAMP/PKA, and MAPK signaling pathways to modulate conidial development, appressorium formation, and pathogenicity [42,43]. Our prior research indicated that the Hog1-MAPK pathway is involved in the infection structure formation of *A. alternata* on hydrophobic and pear wax extract-coated surfaces [18,44]. However, the reaction of Slt2-MAPK to the pear wax of *A. alternata* needed additional clarification. Consequently, we assessed the gene expression level of the *AaSlt2* gene on surfaces coated with hydrophobic and pear wax extract. The findings indicated that the expression level of *AaSlt2* was markedly up-regulated during infection structure formation in *A. alternata* on a surface coated with pear wax extract (Figure 3). Moreover, experiments on onion epidermis indicated that the absence of *AaSlt2* reduced spore germination, appressorium formation, and infection hyphae formation rates (Figure 4), implying that *AaSlt2* is crucial for infection structure formation in *A. alternata* when identifying and reacting to pear wax. Likewise, in *Magnaporthe oryzae* [45], *B. cinerea* [46] and *S. sclerotiorum* [41], the loss of *Slt2* impacted the production of infection structures.

The Slt2-MAPK pathway plays a conserved role in maintaining cell wall integrity. Congo red and SDS impede the polymerization reaction of chitin, thereby damaging the cell wall [47]. Various exogenous chemicals were introduced to the PDA for investigation. The results indicated that growth inhibition was more pronounced in Δ*AaSlt2* compared with that in WT (Figure 5), suggesting that *AaSlt2* plays a vital role in preserving cell wall integrity and adapting to oxidative stress in *A. alternata*, which exhibited similarities to prior research on *G. lucidum* [22]. Nonetheless, in *B. cinerea*, the absence of *Slt2* did not influence protoplast release or cell wall integrity, suggesting that *Slt2* serves distinct roles in maintaining cell wall integrity across various fungi. As a result, certain pathways may be elucidated using transcriptome and metabolomics research.

The Slt2-MAPK pathway also plays a conserved role in pathogenicity. Infection experiments demonstrated that the deletion of *AaSlt2* markedly reduced pathogenicity (Figure 6). Similarly, the deletion of *Slt2* markedly reduced pathogenicity in *P. sojae* [21], *B. bassiana* [48], and *Cryphonectria parasitica* [49]. Fungal virulence determinants comprise CWDEs, effector proteins, and toxins. Pathogens can secrete CWDEs to breakdown the host plant’s cell wall during infection. We assessed the activity of CWDEs. The findings indicated a considerable reduction in PG, PMG, Cx, and β-glucosidase activity in the Δ*AaSlt2* strain (Figure 7).

Melanin included in the fungal cell wall safeguards fungal pathogens against environmental stressors. Nonetheless, melanin functioned as a harmful component in fungal pathogens [50]. *A. alternata* can synthesize melanin, which imparts color to its colonies. Upon inoculation on PDA, it first develops white mycelium, and the colony color transitions from grey to olive or olive-brown over time [51]. In this scenario, melanin accumulation was markedly reduced in the Δ*AaSlt2* strain compared to the wild type (Figure 8A), consistent with prior research on *Cochliobolus heterostrophus* [52] and citrus *A. alternata* [19]. Melanin production, facilitated by the MAPK cascade pathway, is linked with virulence in pear *A. alternata*. Li et al. [53] established that the deletion of *AaPKS1* decreased the pathogenicity of *A. alternata*.

During pathogenesis, several *Alternaria* species can produce both host-specific toxins (HSTs) and non-host specific toxins (NHST). The predominant category of toxins linked to *A. alternata* JT-03 comprises ALT and TEN. ALT and TEN are non-host specific toxins that result in numerous food safety issues and endanger human health. The elimination of *AaSlt2* dramatically lowered ALT and TEN content (Figure 8B), suggesting that *AaSlt2* is involved in the toxin synthesis process. In *Fusarium graminearum*, the ablation of *Slt2* reduced toxin levels [54]. However, the precise processes of *AaSlt2* in mycotoxin generation require additional investigation through the study of toxin-encoding genes by molecular methods. 

## 5. Conclusions

In conclusion, we successfully engineered *AaSlt2* mutant (Δ*AaSlt2*) and complementary strains (Δ*AaSlt2*-C), demonstrating that *AaSlt2* regulates spore morphology, vegetative growth, and environmental stress adaptation in *A. alternata*. The current work has shown that *AaSlt2* is essential to infection structure formation and the virulence of *A. alternata*. The findings enhance our understanding of the molecular mechanisms of *A. alternata* infection and establish a theoretical foundation for the development of target-specific fungicides for postharvest disease management.

## Figures and Tables

**Figure 1 jof-10-00774-f001:**
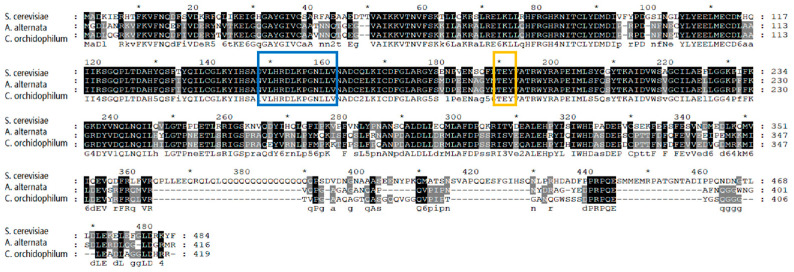
Amino acid sequence alignments of AaSlt2 with homologous protein sequences of *Saccharomyces cerevisiae* (NP_011895.1) and *Colletotrichum orchidophilum* (XP_022474739). The orange box indicates the phosphorylation motif and the blue box indicates the characteristic sequences.

**Figure 2 jof-10-00774-f002:**
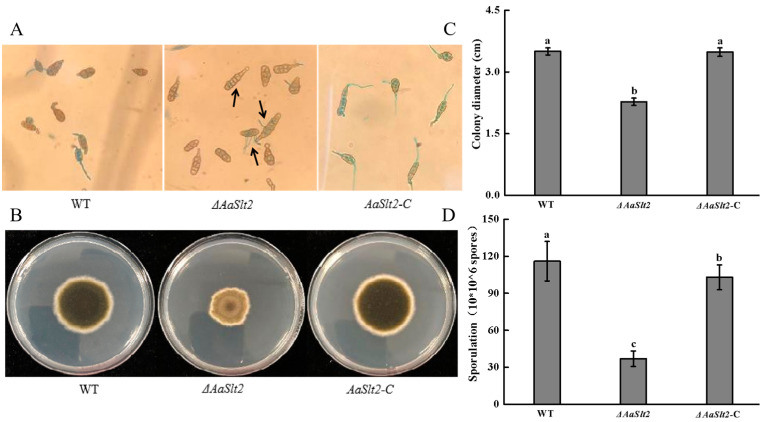
*AaSlt2* is required for vegetative growth and spore development of *A. alternata*. (**A**) Spore development of the indicated strains on PDA medium. The black arrow indicates the abnormal spore of *A. alternata*. (**B**) 5-day-old PDA medium and colony morphology of the WT, Δ*AaSlt2*, and Δ*AaSlt2*-C at 28 °C. (**C**) Colony diameter of WT, Δ*AaSlt2*, and Δ*AaSlt2*-C on PDA medium 5 days after incubation at 28 °C. (**D**) Sporulation of WT, Δ*AaSlt2*, and Δ*AaSlt2*-C on PDA medium 5 days after incubation at 28 °C. Bars indicate standard error (±SE). Different letters on the bars for each treatment indicate significant differences at *p* < 0.05 by Duncan’s multiple range test.

**Figure 3 jof-10-00774-f003:**
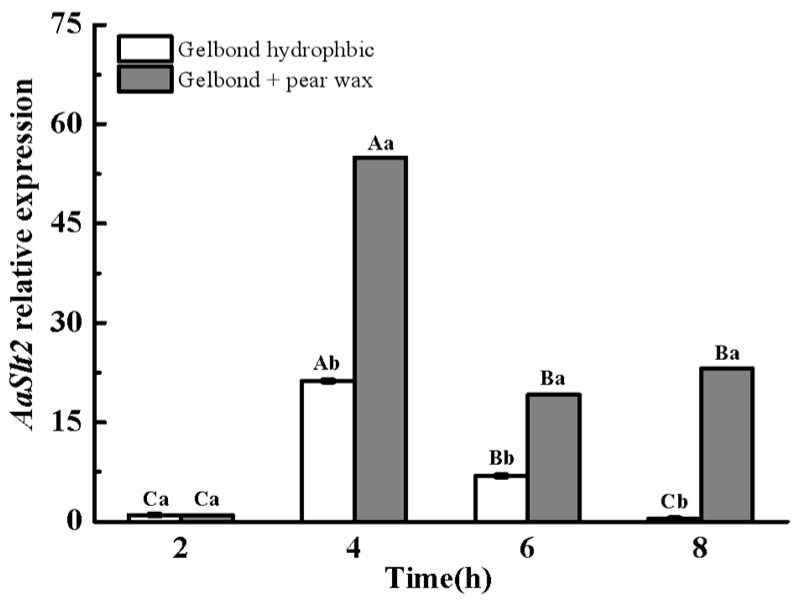
Relative expression levels of *AaSlt2* during infection structural differentiation of *A. alternata* on hydrophobic and pear wax extract-coated surface. Intra-group comparisons were made to compare the changes in *AaSlt2* gene expression during infection structural differentiation of *A. alternata* under hydrophobic and pear wax substrates, respectively. Extra-group were made to compare the changes in *AaSlt2* gene expression under hydrophobic and pear wax substrates at the same stage. Vertical lines indicate the standard error (±SE) of the means. Capital letters indicate intra-group differences. Lowercase letters indicate extra-group differences. Different letters on the bars for each treatment indicate significant differences at *p* < 0.05 by Duncan’s multiple range test.

**Figure 4 jof-10-00774-f004:**
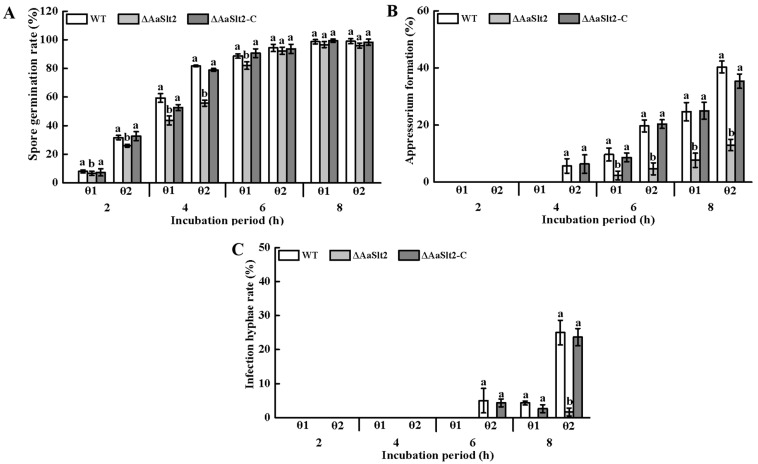
*AaSlt2* regulates the infection structure differentiation of *A. alternata* in vivo. WT, Δ*AaSlt2*, and Δ*AaSlt2*-C spore suspensions (20 µL) were dripped onto intact onion epidermis (θ1) and pear wax extract-coated onion epidermis (θ2) with three replicates and incubated at 28 °C. The percentages of spore germination, appressorium formation, and infection hyphae formation were calculated under a microscope at 2, 4, 6 and 8 h after incubation. The spore germination rate, appressorium formation rate, and infection hyphae formation rate were obtained by counting the number of germinated spores, appressorium formation, and infection hyphae formation in 100 spores and multiplying the result by 100%. (**A**) The spore germination rate of *A. alternata*. (**B**) The appressorium formation rate of *A. alternata*. (**C**) The infection hyphae formation rate of *A. alternata*. Vertical lines indicate standard error (±SE). Different letters on the bars for each treatment indicate significant differences at *p* < 0.05 by Duncan’s multiple range test.

**Figure 5 jof-10-00774-f005:**
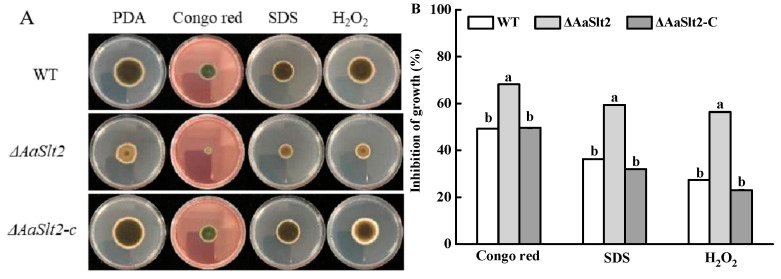
Colony morphology (**A**) and inhibition of growth (**B**) of WT, Δ*AaSlt2,* and Δ*AaSlt2*-C strains on PDA with different exogenous compounds. Lesions were measured at 5 days. Inhibition of growth (%) = (x − y)/x, where x is the colony diameter of WT, Δ*AaSlt2* and Δ*AaSlt2*-C strains on PDA, y is the colony diameter of WT, Δ*AaSlt2* and Δ*AaSlt2*-C strains on PDA with different exogenous compounds. Means and standard deviations were calculated from three replicates. Vertical lines indicate standard error (±SE) of the means. Different letters on the bars for each treatment indicate significant differences at *p* < 0.05 by Duncan’s multiple range test.

**Figure 6 jof-10-00774-f006:**
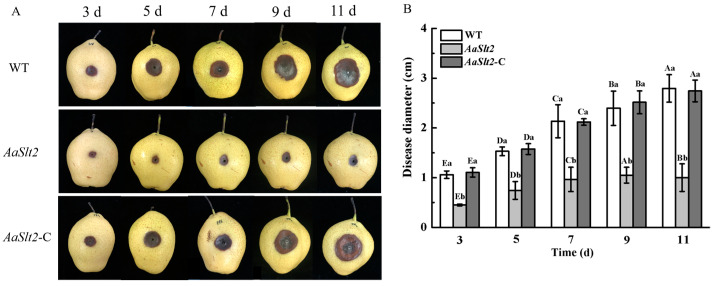
The lesion morphology (**A**) and disease diameter (**B**). Lesions were measured at 3, 5, 7, 9 and 11 days, respectively. Means and standard deviations were calculated from three replicates. Vertical lines indicate standard error (±SE) of the means. Intra-group comparisons were made to compare changes in the same treatment (inoculation with WT/Δ*AaSlt2*/Δ*AaSlt2*-C) at different incubation times (3, 5, 7, 9 and 11 days). Extra-group were made to compare the changes of differences treatments (inoculation with WT, Δ*AaSlt2* and Δ*AaSlt2*-C) at the same incubation times. Capital letters indicate intra-group differences. Lowercase letters indicate extra-group differences. Different letters on the bars for each treatment indicate significant differences at *p* < 0.05 by Duncan’s multiple range test.

**Figure 7 jof-10-00774-f007:**
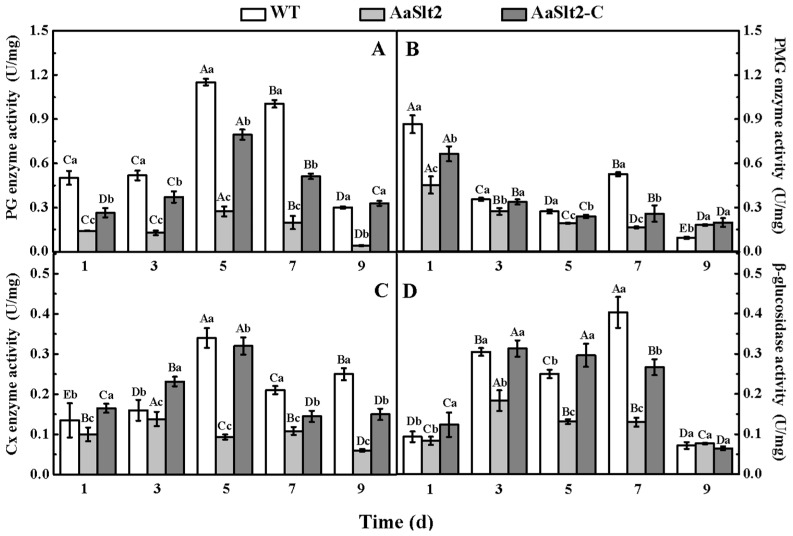
PG (**A**), PMG (**B**), Cx (**C**) and β-glucosidase (**D**) activity of WT, Δ*AaSlt2* and Δ*AaSlt2*-C. Bars indicate standard error (±SE). Intra-group comparisons were made to compare changes in the same treatment (inoculation with WT/Δ*AaSlt2*/Δ*AaSlt2*-C) at different incubation times (3, 5, 7, 9 and 11 days). Extra-group were made to compare the changes of differences treatments (inoculation with WT, Δ*AaSlt2* and Δ*AaSlt2*-C) at the same incubation times. Capital letters indicate intra-group differences. Lowercase letters indicate extra-group differences. Different letters on the bars for each treatment indicate significant differences at *p* < 0.05 by Duncan’s multiple range test.

**Figure 8 jof-10-00774-f008:**
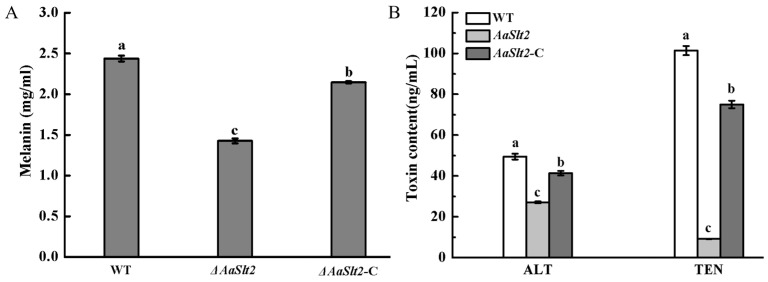
*A. alternata* melanin content (**A**) and toxin content (**B**). Bars indicate standard error (±SE) of three replicates. Different letters on the bars for each treatment indicate significant differences at *p* < 0.05 by Duncan’s multiple range test.

## Data Availability

The original contributions presented in the study are included in the article/Appendix A, further inquiries can be directed to the corresponding author.

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
