# Peer review of "AaSlt2* Is Required for Vegetative Growth, Stress Adaption, Infection Structure Formation, and Virulence in *Alternaria alternata"

_jof, 2024, doi:10.3390/jof10110774_

Round 1
Reviewer 1 Report (Previous Reviewer 1)
I believe the manuscript has been sufficiently improved and now warrants publication in jof
I believe the manuscript has been sufficiently improved and now warrants publication in jof
Reviewer 2 Report (Previous Reviewer 2)
I am happy with the modifications incorporated.
I have no further comments.
This manuscript is a resubmission of an earlier submission. The following is a list of the peer review reports and author responses from that submission.
Round 1
Reviewer 1 Report
Although the results presented in this manuscript is not suprising, it provided extra evident to support the important role of Slt2-MAKP pathway in an important plant pathogen.
Below are the major comments.
1) The author could conduct some basic growth profiles similar as Fig. 6 to verify whether the deletion of Slt2 could affect the fungal utilization of different polysaccharides of plant cell wall, such as grown on PG or cellulose. This could confirm the result in Fig. 8 that Slt2 is involved in control of CWDEs.
2) In the discussion part, you mentioned that the further investigation is need to better understanding of the mechanisms of Slt2. A bit more detailes need to be added here, such as additional RNA-seq or phospho-proteomics analysis.
3) In lengeds of Figures/tables of statistical comparison (Fig. 5, 7, 8, 9 Table. 1 ), what are the exact meaning about "different letters indicate statistical differences". This is a bit confusing.
4) Some abbreviations have not been well explained or descibed, such as PMG, Cx. Legend of Fig. 1, the "TEY" is a phosphyrylation motif, instead of sites. Which site(s) are the exact modified position, T or Y?
See above major comments
Author Response
|
Comments 1: The author could conduct some basic growth profiles similar as Fig. 6 to verify whether the deletion of Slt2 could affect the fungal utilization of different polysaccharides of plant cell wall, such as grown on PG or cellulose. This could confirm the result in Fig. 8 that Slt2 is involved in control of CWDEs. |
|
Response 1: Thank you for pointing this out. We agree with this comment. We will do further research on the cell wall and will add this section in subsequent experiments. |
|
Comments 2: In the discussion part, you mentioned that the further investigation is need to better understanding of the mechanisms of Slt2. A bit more detailes need to be added here, such as additional RNA-seq or phospho-proteomics analysis. |
|
Response 2: Thank you for pointing this out. We agree with this comment. Therefore, we have added the related discussion about further investigating the mechanisms of Slt2 response to oxidative stress and CWI through transcriptomics and metabolomics in Line 429. Besides, the related content about clarifying the mechanism of Slt2 regulation of A. alternata toxin through analyzing toxin synthesis genes have been also added in Line 456. |
|
Comments 3: In lengeds of Figures/tables of statistical comparison (Fig. 5, 7, 8, 9 Table. 1 ), what are the exact meaning about "different letters indicate statistical differences". This is a bit confusing. |
|
Response 3: Thank you for pointing this out. We have carefully considered your suggestions and thought that our statement is a bit confusing. Therefore, we corrected this sentence in lengeds of Figures 3, 4, 5, 6, 7, 8, 9. |
|
Comments 4: Some abbreviations have not been well explained or descibed, such as PMG, Cx. Legend of Fig. 1, the "TEY" is a phosphyrylation motif, instead of sites. Which site(s) are the exact modified position, T or Y? |
|
Response 4: Thank you for pointing this out. We agree with you. Therefore, we add the definitions of PG (polygalacturonase), PMG (pectinmethylgalacturnase) and Cx (cellulase) in Line 172, 186. In legend of Fig. 1, the "TEY" is a phosphyrylation motif, the specific modification sites are currently unknown, we revised this word in Line 244. |
|
4. Response to Comments on the Quality of English Language |
|
Point 1: Moderate editing of English language required. |
|
Response: The language of the manuscript has been improved as suggested. The revision is marked in green in our revised manuscript. |

Reviewer 2 Report
The current study unravels the role of Slt2 in growth and development of A. alternata. Several of the experiments reveal the importance of Slt2 in A. alternata disease development, however I have a few concerns over the over speculative comments and use of grammar throughout the manuscript.
I have few major issues for the manuscript which are as follows:
1. The manuscript contains numerous spelling errors and grammatical mistakes. I have highlighted a few but there are many, i would suggest careful reading and analysis by a proficient language editor.
2. The current study clearly shows Slt2 deletion mutants are retarded in A. alternata germination and growth under PDA, so is it it evident that all the toxicity, enzyme and infection assays performed would have reduced mycelia growing which would directly impact the overall results. Hence to claim it performs such diverse activities is an overstatement and needs to be modified throughout the manuscript. Slt2 mutant shows reduced growth under PDA media so it is obvious there will be difference under cell wall inhibitors and oxidizing agents as well.
3. In several instances a percentage has been used to claim the increase or decrease in activity of Slt2 mutant in comparison to wild type. It would be better to include in the supplementary file a detailed analysis of what all the comparison values were.
4. The abstract needs to be rewritten, lines like 'was clarified through targeted gene disruption' and overuse of words like further, meanwhile, moreover needs to be minimized.
5. throughout the manuscript the role of Slt2 in MAPK has not been shown. Although it has been claimed to have role in several activities, it is an overstatement to be involved in kinase activity without any validation.
6. Methods section requires more elaboration and detailed description instead of only citing to previous references.
7. In Figure 2, Controls showing empty vector only GFP is missing. Why does entire mycelial strand strain with DAPI stain?
8. Line 350-351: 'hypersensitive to Congo red, SDS and H2O2': how was hypersensitive measured? Does lack of growth under these conditions mean they are hypersensitive?
Overall the manuscript clearly shows the importance of Slt2 role in Alternatia's growth and virulence. However some of the claims are too far fetched and more analysis needs to be done. I would advice to tone it to a suggestive level rather than claiming it to perform all these functions.
Author Response
|
3. Point-by-point response to Comments and Suggestions for Authors |
|
Comments 1: The manuscript contains numerous spelling errors and grammatical mistakes. I have highlighted a few but there are many, i would suggest careful reading and analysis by a proficient language editor. |
|
Response 1: Thank you for pointing this out. The language of the manuscript has been improved as suggested. The revision is marked in green in our revised manuscript. |
|
Comments 2: The current study clearly shows Slt2 deletion mutants are retarded in A. alternata germination and growth under PDA, so is it it evident that all the toxicity, enzyme and infection assays performed would have reduced mycelia growing which would directly impact the overall results. Hence to claim it performs such diverse activities is an overstatement and needs to be modified throughout the manuscript. Slt2 mutant shows reduced growth under PDA media so it is obvious there will be difference under cell wall inhibitors and oxidizing agents as well. |
|
Response 2: Thank you for pointing this out. We agree with this comment. The growth inhibition rates of WT, ∆AaSlt2, and ∆AaSlt2-C strains were compared, and suggested that AaSlt2 is involved in oxidative and cell wall inhibitors stress. |
|
Comments 3: In several instances a percentage has been used to claim the increase or decrease in activity of Slt2 mutant in comparison to wild type. It would be better to include in the supplementary file a detailed analysis of what all the comparison values were. |
|
Response 3: Thank you for pointing this out. We agree with this comment. In Figure 5, spore germination rate, appressorium formation rate and infection hyphae formation rate were obtained by counting the number of germinated spores, appressorium formation and infection hyphae formation in 100 spores and multiplying the result by 100%. In Figure 6B, inhibition of growth (%) = (x-y)/x, x: colony diameter of WT, ∆AaSlt2 and ∆AaSlt2-C strains on PDA, y: colony diameter of WT, ∆AaSlt2 and ∆AaSlt2-C strains on PDA with different exogenous compounds. |
|
Comments 4: The abstract needs to be rewritten, lines like 'was clarified through targeted gene disruption' and overuse of words like further, meanwhile, moreover needs to be minimized. |
|
Response 4: Thank you for pointing this out. We agree with this comment. Therefore, we revised the abstract correspondingly. |
|
Comments 5: throughout the manuscript the role of Slt2 in MAPK has not been shown. Although it has been claimed to have role in several activities, it is an overstatement to be involved in kinase activity without any validation. |
|
Response 5: Thank you for pointing this out. We agree with this comment. Based on the current study, we will follow up with further experiments to further elucidate the specific role of Slt2 in MAPK. |
|
Comments 6: Methods section requires more elaboration and detailed description instead of only citing to previous references. |
|
Response 6: Thank you for pointing this out. We agree with this comment. Therefore, we supplemented the detailed methodology in Methods section. |
|
Comments 7: In Figure 2, Controls showing empty vector only GFP is missing. Why does entire mycelial strand strain with DAPI stain? |
|
Response 7: Thank you for pointing this out. We agree with this comment. Fungal hyphae contain nucleus, which will be stained blue when stained with cytoplasmic dye DAPI. Preliminary software prediction showed that Slt2 was located in the cytoplasmic nucleus, so the co-localization of green and blue fluorescence indicated that Slt2 was located in the cytoplasmic nucleus. |
|
Comments 8: Line 350-351: 'hypersensitive to Congo red, SDS and H2O2': how was hypersensitive measured? Does lack of growth under these conditions mean they are hypersensitive? |
|
Response 8: Thank you for pointing this out. We agree with this comment. The ‘hypersensitive’ is exaggerated, so we revised the results Line 315. |
|
4. Response to Comments on the Quality of English Language |
|
Point 1: Extensive editing of English language required. |
|
Response: The language of the manuscript has been improved as suggested. The revision is marked in green in our revised manuscript. |

Reviewer 3 Report
Qianqian Jiang and coauthors present their work on the molecular characterization of the gene Slt2 in Alternaria alternata. Such work is essential to the characterization and validation of genes.
Despite my numerous critics of the methods, the work done is noteworthy. The work done is a contribution to the field but it need to be better communicated. For example, explain why you use pear-wax coating on gelbond films and onions. What is the composition of the pear-wax? What components are known to generate a response in Alternaria?
line 14: “Deletion of Aaslt2 was defective”: the organism with the deletion (a mutant) can be defective but not the deletion. Please rephrase.
Line 18: AaSlt2 deletion mutant and then next line DAaSlt2 strain. Please homogenize the how you refer to mutant.
Line 30: precise what are AK and ACT abbreviations
Line 43: secondary metabolism, please correct
Line 53: explain what are hog1, slt2, smk1, fus3 and kss1. Are they different proteins ? In what do they differ?
Develop further the introduction by transferring line 311-315 to the introduction.
LINE 69: describe the inhibitor.
Line 168: Describe the PK-c superfamily
Line 167-174: What is the percentage of sequence homology?
Line 184-186: More septa than WT. Provide number. Is it a significant difference?
Line 216, 275: What are "extra-group difference"? Are the statistical test done within or across treatments?
Line 233-238: What are infection hyphae?
Table 1: This in not readable. The data is presented as inhibitory rates with positive and negative values. What does it mean? Is it calculated as WT minus Treatment? the description of the data analysis is missing Does a positive value mean that the colony is larger or smaller than the PDA control? I would advice to provide the average colony size including WT. Then do proper statistics.
Line 278 and following: How was that experiment conducted? Is it on coated film? Liquid culture? How were these enzymatic activity measured? What is U/mg? Please describe
Line 195: how was the melanin quantified? Was it in comparison to a standard?
Line 316-321: the % of homology was never provided. Please do it before concluding about conservation and evolution.
332: What are the physiochemical clues studied in this manuscript (e.g. pear-wax)? Before concluded on that, it should be clear what was done and why. What component of the pear-wax elicit the response?
375: The production of toxins should be described in the introdution. Is it optional? Is it host specific?
377: How, Why were to level of toxic lower? Please explain
Author Response
Line 77: "The wild-type strain was preserved in this laboratory". What is the method? Are the spores/hyphae preserved, in what conditions, temperatures? It is standard to have spores suspended in glycerol and maintained frozen. Is it the case??
Response: Thank you for pointing this out. The specific preservation method of the strain is supplemented in Line 84. We preserved spore suspension to 80% glycerol (3:1, v/v) and stored at -80℃.
Line 85: "Conserved domains of AaSlt2 was predicted by Conserved domains". What is the methodology behind the function "conserved domains" in the software?
Response: Thank you for pointing this out. This software identifies conserved structural domains of genes by analyzing and comparing amino acid sequences of different species.
Line 93: What is AtMT transformation? Where single spores selected?
Response: Thank you for pointing this out. AtMT (agrobacterium tumefaciens-mediated transformation) is to insert the target gene onto the T-DNA of Ti plasmid, and through the transformation effect of agrobacterium, the target gene is introduced into the fungal cell and inserted into the DNA of the chromosome in the fungal cell, so that the genetic characteristics of the target gene can be maintained and expressed stably. AtMT is becoming a popular effective system as an insertional mutagenesis tool in filamentous fungi. The transformants were selected from induction medium plates containing 200 μM acetosyringone and were verified using PDA plates containing carbeniclin (500 μg/ml) and hygromycin B (250 μg/ml). We supplemented relevant content in Line 102.
Line 105: "The AaSlt2-C strains [plurals, so how many strains?] was used to observe subcellular localization? Localization of what?
Response: Thank you for pointing this out. We obtained multiple complementation strains, and we selected one of them for subsequent experiments as well as subcellular localization. The Slt2 gene was previously localized in the cytoplasmic nucleus using the WoLF PSORT software, so we further stained the spores at different developmental stages (2h, 4h, 6h) using the cytosolic dye DAPI (4',6-diamidino-2-phenylindole), which showed co-localization of green and blue fluorescence, suggesting that the Slt2 gene function is indeed in the nucleus.
Line 108: What is DAPI?
Response: Thank you for pointing this out. DAPI (4',6-diamidino-2-phenylindole) is a fluorescent dye that binds strongly to DNA and is commonly used in fluorescence microscopy. DAPI can pass through intact cell membranes and can be used to stain living and fixed cells, and can also assist in determining the location of cell signals. When cells are stained with DAPI, changes in the morphology of the nucleus can be seen under a fluorescent microscope.
Line 119: "Spore suspension of WT were collected with hydrophobic and pear-wax extract coated hydrophobic films or 2,4, 6, 8h". I don't understand, especially the usage of "collected" in the context. Where the spores grown on either hydrophobic or pear-wax extract films for x time? Is it two different treatments?
Response: Thank you for pointing this out. Two different treatments were used in this experiment, treatment 1: hydrophobic film was spread on the slide; treatment 2: hydrophobic film was spread on the slide, and then pear-wax was evenly coated on the hydrophobic film. Under the above two treatments, WT spore suspensions (10^5 spores/mL) were added, and the spore suspensions were scraped from the film after 2, 4, 6 and 8 h, respectively, after centrifuged, the precipitates were used for RNA extraction.
Line 122: please describe in one or two sentences what is the methodology in addition to the reference
Response: Thank you for pointing this out.
We supplemented the sampling method in detail. The expression level of the target gene was calculated using the 2-∆∆CT method (Livak and Schmittgen). We supplemented relevant content in Line 131-136.
Line 124: How was the onion coated with pear-wax? Please describe the method in a few words.
Response: Thank you for pointing this out. The inner membrane of the onion epidermis was excised and sectioned into 20 × 20 mm squares, which were subsequently positioned on slides. Pear wax, dissolved in chloroform, was uniformly applied to the inner membrane of the onion with an applicator. We supplemented relevant content in Line 139.
Line 132: "The media ... were used to induced spore suspensions". This doesn't make any sense. From the images, plates containing media xyz where inoculated with spore suspension.
Response: Thank you for pointing this out. We description of the methods is confusing, so we added specific methods in Line 149-152.
Line 133: How were the colony size estimated? What is the precision of the methodolgy you used? Is it based on images? Estimation of the diameter of the colony?
Response: Thank you for pointing this out. The colony size is measured using a ruler through a cross-over method. Measurement with two significant digits. It is directly measured on a plate, not based on images. The data are actual data measured directly and are not an estimate.
Line 140: What is a bagat?
Response: Thank you for pointing this out. Our word choice may not be accurate. Therefore, we use ‘bag’ to replace ‘bagat’ to make the description more accurate in Line 160.
Line 144: Please describe the methods in one or two sentences.
Response: Thank you for pointing this out. We supplemented the measurement methods of cell wall degrading enzyme activity in Line 172-188.
Line 151: Is it a absorbance based methodology? HPLC? Paper based? Please describe the method
Response: Thank you for pointing this out. It is a absorbance based methodology, and the detailed method were supplemented in Line 191-206.
Line 156: "the samples of the toxins were consistent with the above". What does it mean? Did you also boiled the samples in NaOH before the acetonitrile extraction?
Response: Thank you for pointing this out. There may be an error in the way it is written here. The toxin samples have not been treated with NaOH. We added detailed methods in Line 208-217.
Line 159: "The subsequent precautions were followed of Xu et al." What precautions? What were the analytical instruments used for the HPLC? What was the software for data analysis? What was the quantification method? What type of array was used [DAD, FAD, Mass spectrometry]?
Response: Thank you for pointing this out.
The parameters of HPLC refer to the method of Xu et al. The analytical instruments is Agilent 1260 Q-TOF mass spectrometer (Agilent, Santa Clara, CA, USA). We used SPSS 18.0 software for data analysis and Origin 8.5 for graphing. We calculate the content of ALT and TEN in a sample by pre-establishing a standard curve of the Altenuene (ALT) and Tentoxin (TEN) standard product. The type of array was used is DAD. We supplemented in Line 208-217.
Line 164: Statistics are only described by "Duncan's method". Never heard of it. I would expect at least a t-test or an anova.
Response: Thank you for pointing this out.
We revised the method of data analysis in Line 219-223.
|
3. Point-by-point response to Comments and Suggestions for Authors |
|
Comments 1: line 14: “Deletion of Aaslt2 was defective”: the organism with the deletion (a mutant) can be defective but not the deletion. Please rephrase. |
|
Response 1: Thank you for pointing this out. We agree with this comment. We revised in Line 15. |
|
Comments 2: Line 18: AaSlt2 deletion mutant and then next line DAaSlt2 strain. Please homogenize the how you refer to mutant. |
|
Response 2: Thank you for pointing this out. We have revised in Line 19, 22. |
|
Comments 3: Line 30: precise what are AK and ACT abbreviations |
|
Response 3: Thank you for pointing this out. AK and ACT toxins are named according to the Alternaria kikuchiana and the tangerine pathotype of Alternaria alternata, so there is no question of abbreviation. |
|
Comments 4: Line 43: secondary metabolism, please correct |
|
Response 4: Thank you for pointing this out. We revised the word in Line 45. |
|
Comments 5: Line 53: explain what are hog1, slt2, smk1, fus3 and kss1. Are they different proteins ? In what do they differ? hog1, slt2, smk1, fus3 and kss1 Develop further the introduction by transferring line 311-315 to the introduction. |
|
Response 5: Thank you for pointing this out. We supplemented the definitions of these five proteins in Line 50. Hog1, slt2, smk1, fus3 and kss1 are different five proteins on different five regulatory pathways on the MAPK pathway, and responsible for regulating different signaling pathways. I'm sorry, slt2, Fus3 and Kss1 are defined as such, so we have nothing to add. |
|
Comments 6: Line 69: describe the inhibitor. |
|
Response 6: Thank you for pointing this out. SB203580 is a specific inhibitor of the MAPK pathway, and the use of this drug will facilitate more in-depth study of the function and mechanism of this pathway. |
|
Comments 7: Line 168: Describe the PK-c superfamily |
|
Response 7: Thank you for pointing this out. The PKc_like superfamily mostly consists of the catalytic domains of serine/threonine-specific and tyrosine-specific protein kinases. It also encompasses RIO kinases, which are characteristic serine protein kinases, aminoglycoside phosphotransferases, and choline kinases. These proteins catalyze the transfer of the gamma-phosphoryl group from ATP to hydroxyl groups in specific substrates such as serine, threonine, or tyrosine residues of proteins. We supplement the definition of the PKc-superfamily in Line 228-234. |
|
Comments 8: Line 167-174: What is the percentage of sequence homology? |
|
Response 8: Thank you for pointing this out. The sequence homology of these three species is 69.28%. |
|
Comments 9: Line 184-186: More septa than WT. Provide number. Is it a significant difference? |
|
Response 9: Thank you for pointing this out. The special spores (more septa) in the ∆AaSlt2 strain accounted for 50 %, so we believe that the spore development of the ∆AaSlt2 strain and the wild type is significant different. |
|
Comments 10: Line 216, 275: What are "extra-group difference"? Are the statistical test done within or across treatments? |
|
Response 10: Thank you for pointing this out. The "extra-group difference" means across treatments. We compared differences of intra-group and extra-group, separately. Intra-group comparisons were made to compare the changes in AaSlt2 gene expression during infection structural differentiation of A. alternata under hydrophobic and pear wax substrates. Extra-group were made to compare changes in AaSlt2 gene expression under hydrophobic and pear wax substrates at the same stage. |
|
Comments 11: Line 233-238: What are infection hyphae? |
|
Response 11: Thank you for pointing this out. The process of infection structural differentiation of A. alternata was consists of four stages. Firstly, the spore germination extends the germ tube, the germ tube continues to elongate at the tip to form appressorium, with the prolongation of time, the tip of the appressorium will extend a bacterial filament (infection hyphae), which will infest the host and make the host sick. |
|
Comments 12: Table 1: This in not readable. The data is presented as inhibitory rates with positive and negative values. What does it mean? Is it calculated as WT minus Treatment? the description of the data analysis is missing. Does a positive value mean that the colony is larger or smaller than the PDA control? I would advice to provide the average colony size including WT. Then do proper statistics. |
|
Response 12: Thank you for pointing this out. We have thought about the results and re-presented them as a Figure. The inhibition of growth (%) = (x-y)/x, x: colony diameter of WT, ∆AaSlt2, and ∆AaSlt2-C strains on PDA, y: colony diameter of WT, ∆AaSlt2, and ∆AaSlt2-C strains on PDA with different exogenous compounds. We revised result in Line 315 and added data in Figure 6B. |
|
Comments 13: Line 278 and following: How was that experiment conducted? Is it on coated film? Liquid culture? How were these enzymatic activity measured? What is U/mg? Please describe |
|
Response 13: Thank you for pointing this out. Mycelium of WT, ∆AaSlt2, and ∆AaSlt2-C were cultured in PDB (culture medium) for 4 days. The enzyme activity unit is defined as the quantity of enzyme necessary to facilitate the liberation of 1 μg of reducing sugar from the substrate by 1 mg of enzyme protein per hour at 37℃. The enzymatic activity measured was supplemented in Line 172-188. |
|
Comments 14: Line 195: how was the melanin quantified? Was it in comparison to a standard? |
|
Response 14: Thank you for pointing this out. Melanin content was determined by spectrophotometer and calculated by substituting into the standard curve. The formula for determining melanin content was: y = x + 0.111/0.791 (where x is the absorbance value). We supplemented our method in Line 190-206. |
|
Comments 15: Line 316-321: the % of homology was never provided. Please do it before concluding about conservation and evolution. |
|
Response 15: Thank you for pointing this out. The sequence homology of these three species is 69.28% and We supplemented in Line 385. |
|
Comments 16: 332: What are the physiochemical clues studied in this manuscript (e.g. pear-wax)? Before concluded on that, it should be clear what was done and why. What component of the pear-wax elicit the response? |
|
Response 16: Thank you for pointing this out. The physiochemical clues is hydrophobic and pear wax in our study. Pear wax is extracted from the epidermis of pears fruit. Our previous studies showed that pear wax can induce the infection structural differentiation of A. alternata. However, we don’t know which component of the pear wax is responsible for this and further research is needed. |
|
Comments 17: 375: The production of toxins should be described in the introdution. Is it optional? Is it host specific? |
|
Response 17: Thank you for pointing this out. A. alternata can produce both host-specific toxins (HSTs) and non-host specific toxin (NHST), and our studies have focused on the NHST. We supplemented in Line 452. |
|
Comments 18: 377: How, Why were to level of toxic lower? Please explain |
|
Response 18: Thank you for pointing this out. The decrease of toxin content in ∆AaSlt2 strain compared to WT suggests that AaSlt2 is involved in the toxin synthesis process, but its specific mechanism needs further investigation. |
|
4. Response to Comments on the Quality of English Language |
|
Point 1: Extensive editing of English language required. |
|
Response : The language of the manuscript has been improved as suggested. The revision is marked in green in our revised manuscript. |

Round 2
Reviewer 1 Report
The authors have addressed my question and I have no further comments
The authors have addressed my question and I have no further comments
Reviewer 2 Report
Out of all the queries raised most of them were not incorporated in the manuscript. It is important that the queries be addressed in the manuscript file instead of replying as response as well as it would raise the readability and quality of the work presented. Regarding comment 2 of first revision, the explanation and changes need to be incorporated in the manuscript.
Referring to comment 7, In Figure 2, Controls showing empty vector only GFP is missing. Why does entire mycelial strand strain with DAPI stain. I am not convinced by what the authors meant to say cytoplasmic nucleus. DAPI is an exclusive nuclear stain and not cytoplasmic hence mycelial staining of cytoplasm can be either due to overstraining or overexposure to DAPI filter.
Reviewer 3 Report
The manuscript is better than the original version. Unfortunately, many of my concerns were not corrected, or answered in the reviewer answer but not in the manuscript where it would benefit to the broad range of readers.
It is better than in the previous version, but there are still a lot of methods that wouldn't be reproducible.
Here are a few examples (not exhaustive):
Line 87-88: why mention ALT and TEN here rather than at line 210?
Line 94: Please describe what is conserved domain in the manuscript, not only in your response to me.
Line 135: "RNA and cDNA were acquired using the manufacturer;'s protocol". Well, at least could you list the kit and the manufacturer?